# REINFORCED IMITATION LEARNING FROM OBSERVATIONS

## ABSTRACT

Imitation learning is an effective alternative approach to learn a policy when the reward function is sparse. In this paper, we consider a challenging setting where an agent has access to a sparse reward function and state-only expert observations. We propose a method which gradually balances between the imitation learning cost and the reinforcement learning objective. Built upon an existing imitation learning method, our approach works with state-only observations. We show, through navigation scenarios, that (i) an agent is able to efficiently leverage sparse rewards to outperform standard state-only imitation learning, (ii) it can learn a policy even when learner's actions are different from the expert, and (iii) the performance of the agent is not bounded by that of the expert due to the optimized usage of sparse rewards.

## 1 INTRODUCTION

Learning by imitating is one of the most fundamental forms of learning in nature (McFarland, 1999; Jones, 2009). Its critical role in cognitive development is also supported by the fact that human brains have special structures, such as the mirror neurons, which are presumed to support this ability (Heyes, 2010). Due to this significance, it has also played a key role in machine learning and robotics (Pomerleau, 1989; Ratliff et al., 2007), especially for the problems where reinforcement learning (RL) can easily be inefficient, *e.g.*, due to the sparsity of the reward signals.

Imagine an infant (the *learner*) observing a caregiver (the *expert*) who is performing a task, *e.g.*, opening a door. From this example, we can derive the following observations on imitation learning (IL). First, unlike the typical imitation learning in machine learning, the true action labels (motor command) executed by the expert is not available to the learner. Although in some cases such as autonomous cars, it may be possible to have access to the internal action labels by deploying special equipment, it is still expensive in general and in many applications not possible at all. Second, the actions that can be executed by the expert and the learner are different because of the differences in body development. This challenge can also easily be raised in real world applications. For instance, we may have a new version of a home robot that needs to learn from demonstrations of the old version which supports only a naive set of actions, *e.g.*, only a subset of the new version. Third, it may be reasonable and realistic to augment imitation learning with sparse reward signals. Even if having access to the labels to every action is unrealistic, in many cases the sparse rewards, *e.g.*, the completion of a task signaled by language or facial expression, can easily and cheaply be obtained. These challenges together with the inherent difficulties of reinforcement learning such as the sparsity of the reward signal and sensitivity to hyperparameter tuning, are required to be dealt with in order to make imitation learning applicable to real and complex challenges.

In this paper, we propose a method for *Reinforced Imitation Learning from Observations* (RILO) to tackle the aforementioned challenges. Following the above observations, the proposed method aims to work efficiently in a setting where (i) labels for expert actions are not available, (ii) a reward signal is only sparsely provided, and (iii) the expert and learner operate in different action spaces. To achieve this, we extend generative adversarial imitation learning (GAIL) (Ho & Ermon, 2016) to improve efficiency for the cases where the expert actions are not available *and* different from the learner actions. The proposed approach can automatically balance learning between imitation and environment rewards. This makes our method learn as fast as imitation learning but also potentially converge to a better policy than the one demonstrated. This is done by gradually, but automatically,

releasing the reliance on the imitation reward and then by learning more from the environment rewards.

Through a series of experiments in simple navigation scenarios (both fully and partially observable), we show that an agent is able to combine both the state observations from an expert and the sparse rewards to achieve better performance than either pure RL or IL with state-only observations. The main contribution of the paper is as follows. By leveraging environment sparse rewards, we propose a method that outperforms standard IL from observations and validate it through a series of systematic experiments. We provide an algorithm that can be applied when expert and learner do not share the same action space. The performance of the learner is not bounded by that of the expert due to suitable use of the environment sparse rewards.

## 2 RELATED WORK

Imitation learning (IL) is a common approach to learn a policy from expert demonstrations, *i.e.* sequences of state-action pairs. IL includes two main categories: (i) behavioural cloning (Bain & Sammut, 1995; Pomerleau, 1989) and (ii) inverse reinforcement learning (Abbeel & Ng, 2004; Ng & Russell, 2000). Behavioral cloning directly learns the mapping from a state to an action by using the true action-labels from demonstrations and thus is a supervised learning method. Inverse reinforcement learning derives a reward function from demonstrations that may be then used to train a policy using the learned reward function.

Recently, Ho & Ermon (2016) proposed GAIL, a method that uses demonstration data by forcing the learner to match state-transition occupancy distribution of the expert using an approach similar to GANs (Goodfellow et al., 2014). Although GAIL is very effective and has attracted research attention (Stadie et al., 2017; Li et al., 2017), it requires expert state-action pairs which are expensive to obtain in many applications.

For this reason, we focus on an IL scenario in which an agent does not use the expensive true action labels from an expert, but uses only the state observations. Following the previous literature, we call this *imitation learning from observation* (ILO). Among few existing works that focus on this problem (Torabi et al., 2018a; Kimura et al., 2018; Stadie et al., 2017; Aytar et al., 2018; Liu et al., 2017), there are two that we find to be the closest to our method (Merel et al., 2017; Torabi et al., 2018b). While both methods are built on top of GAIL, unlike GAIL, they target the case where the state-only observations are provided. Contrary to our approach, these methods work under the pure IL setting, *i.e.* they do not take advantage of the potential availability of sparse rewards. Some other line of works consider state-only expert observations during training but also require expert observations at test time (Pathak et al., 2018; Duan et al., 2017; Borsa et al., 2017).

Another related line of works is to use demonstrations to improve the exploration efficiency of RL under sparse reward settings (Nair et al., 2017; Hester et al., 2017; Vecerík et al., 2017). Unlike us, these works however use the expensive state-action paired demonstrations, and treat them as self-generated. They also consider the optimal demonstrations only. While Kang et al. (2018) and Zhu et al. (2018) do not rely on these assumptions, they still use state-action demonstrations and assume both expert and learners share the same action space. Also, they used the full reward by combining the environment reward and imitation reward as a convex combination whose weights are manually set as a hyperparameter. Our method can, however, automatically adapt the balance.

The last two works that we would like to mention are (Gupta et al., 2017; Gao et al., 2018). The former considers agents that may be morphologically distinct, however their approach assumes that time alignment is trivial which does not hold in our experimental setup. The latter works on imperfect demonstrations which makes the work related to our (considering different actions spaces implies dealing with imperfect demonstrations), however their approach assumes the access to the demonstrations (including expert actions).

## 3 METHOD

Our method is based on the recently proposed GAIL (Ho & Ermon, 2016). This method suggests an adversarial training approach as a way to make the distribution of state-action pairs of the learner as indistinguishable as possible from that of the expert. Although it achieves good performances,

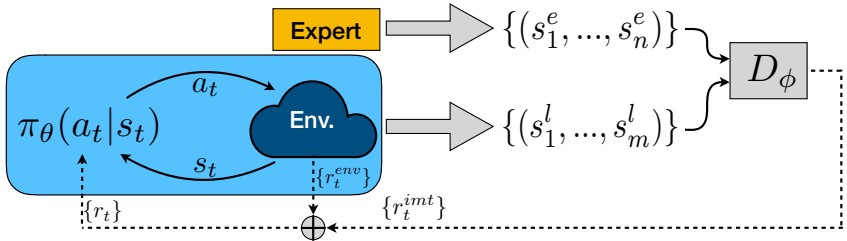

Figure 1: A representation of RILO framework. An agent (learner) with a policy $\pi_\theta$ interact with an environment producing a trajectory of states and a sparse reward. Imitation rewards are acquired by comparing state-only observations from the learner and the expert. The policy is updated with a combination of both rewards.

GAIL relies on two strong assumptions, which we want to overcome in our method: (i) the learner has access to the actions of the expert and (ii) the expert and the learner possess the same action space.

Our setting differs from the standard GAIL approach in three ways. First, we assume that state-only expert observations are provided as a dataset of trajectories $T_e = \{(s_1^i, ..., s_{n_i}^i)\}_{i=1}^N$. These trajectories consist of observations derived by executing an expert policy on a (possibly partially observable) Markov decision process with state space $\mathcal{S}$, action space $\mathcal{A}$, a transition probability $p(s_{t+1}|s_t, a_t)$, a reward function $r$, and discount factor $\gamma$. In our setting, an expert policy $\pi_e(a_t|s_t)$ performs actions in $\mathcal{A}^e \subset \mathcal{A}$. Second, we also consider the learner to have actions in $\mathcal{A}^l \subset \mathcal{A}$, potentially different from $\mathcal{A}^e$. Note that both the expert and the learner operate on the same state space $\mathcal{S}$ and we assume the same transition probability defined on the superset of agents' action spaces, *i.e* $\mathcal{A}^e \cup \mathcal{A}^l \subseteq \mathcal{A}$.

Since the expert and the learner can perform different actions, the two agents may have different optimal policies. In this case, pure imitation learning methods would end up with a learner having a sub-optimal policy. We propose *self-exploration* method to utilize the availability of sparse environment rewards to escape from the sub-optimal policy (see Subsection 3.3).

## 3.1 OVERVIEW

Our approach is composed of two different components (see Figure 1). A *policy* $\pi_\theta : \mathcal{S} \rightarrow \mathcal{A}^l$ outputs an action given its current state. A *discriminator* $D_\phi : \mathcal{S} \times \mathcal{S} \rightarrow [0, 1]$ is responsible for identifying the agent (either expert or learner) that has generated a given pair of states.

Our goal is to optimize the following minimax objective:

$$\min_\theta \ \max_\phi \ \mathbb{E}_{\pi_\theta}[\log(1 - D_\phi(s_i, s_j))] + \mathbb{E}_{T_e}[\log(D_\phi(s_i, s_j))] \,, \tag{1}$$

where $(s_i, s_j)$ is a tuple of states generated by the policy $\pi_\theta$ or $\pi_\phi$. Similar to Zhu et al. (2018), the policy is trained to maximize the discounted sum of the final reward $r_t$, which is the sum of the environment reward $r_t^{env}$ and the imitation reward $r_t^{imt}$ given by the discriminator:

$$r_t = r_t^{env} + \lambda \cdot r_t^{imt} \,, \ \ \lambda > 0 \,. \tag{2}$$

In the next subsection, we describe how we define the imitation reward $r_t^{imt}$. The reward $r_t$ from Equation 2 encourages the learned policy to visit similar paths as the expert, while obtaining high environment reward by achieving the goal. In all our experiment we always used $\lambda = 1$.

Each policy iteration consists of the following steps: (i) collect observations with the current learner policy, (ii) compute the discriminator scores for pairs of states where each pair is from either the expert or the learner, and update the discriminator, and (iii) compute the final rewards $\{r_t\}$ and update the policy. To update the policy, we used the A2C algorithm, a synchronous variant of A3C (Mnih et al., 2016). See Figure 1 for the illustration of this procedure.

For the rest of this section, we describe the two main contributions that we found to be fundamental to make an agent learn to complete a task in the RILO setting: different new imitation rewards *and* a method to efficiently combine state-only observations and sparse rewards.

Table 1: Comparison of different imitation rewards. Each method assumes a different input for the discriminator. GAIL considers action-state pairs and is shown only for reference. The following methods assume, respectively, consecutive pairs of states, a single state only, a pair with a given state and random previous state, and all possible pairs of states containing a given state. Each method is responsible for an imitation reward. In RTGD, $\rho(t)$ returns a random integer smaller than $t$. In ATD, $D^*$ is a clamped version of $D$ that never returns a value smaller than the average of all states.

| | GAIL | CSD | SSD | RTGD | ATD |
|---|---|---|---|---|---|
| **Input** | $(a_t, s_t)$ | $(s_{t-1}, s_t)$ | $(s_t, 0)$ | $(s_{\rho(t)}, s_t)$ | $\{(s_i, s_t) \| i \neq t\}$ |
| **Score $\mu_t$** | $1 - D(a_t, s_t)$ | $\frac{1 - D(s_{t-1}, s_t)}{0.5}$ | $\frac{1 - D(s_t)}{0.5}$ | $\frac{1 - D(s_{\rho(t)}, s_t)}{0.5}$ | $\frac{\sum_{i \neq t}(1 - D^*(s_i, s_t))}{0.5 \cdot (T-1)}$ |
| **Reward $r_t^{imt}$** | | | $-\log \mu_t$ | | |

## 3.2 Imitation Rewards

As mentioned above, contrary to GAIL, we do not take into account state-action pairs but focus solely on state observations. Torabi et al. (2018b) used pairs of consecutive transition states as a proxy to encode unobserved actions from expert. As we shall show in the experiment section, this strategy fails when two agents have different action spaces. In this case, the discriminator can easily discriminate between the expert and the learner, because short-term state transitions provide strong information about the agents' actions. We call this approach Consecutive States Discrimination (CSD).

Other approaches (Merel et al., 2017; Zhu et al., 2018) provide only the current state, simply ignoring the action $a_t$ used in the original GAIL approach. The main limitation of this method is that the discriminator is not aware of the dynamics of the agent move trajectory. As we show later, this approach requires a large amount of expert observations to provide a reasonable performance. We dub this method Single State Discrimination (SSD).

We consider CSD and SSD as our baseline methods and propose two alternatives for the input to the discriminator to circumvent the aforementioned issues.

We call our first method Random Time Gap Discrimination (RTGD). In RTGD, a pair of states is chosen with random time gap. This simple and effective method retains the information about the agent's trajectory dynamics as is in CSD, but avoids the limitation of CSD by not limiting to very short-term transitions, resulting in state-pairs not trivial to the discriminator. Furthermore, we can limit the minimum gap between the pairs so that the possibility of having short-term transition pairs is completely excluded.

Another solution would be to consider all state pairs instead of a single one. In this case, the final reward at time $t$ is based on the discriminator scores of all pairs containing state $s_t$. However, that would still give the discriminator many short-term transitions. A naive way would be to exclude them using a threshold parameter, as done in the RTGD.

Our second method, Averaged per Time Discrimination (ATD), makes a better use of these scores to improve discrimination. First, we compute the mean of the scores of all pairs. Then, the lowest discrimination scores (that is, the pairs in which the discriminator is more confident that it comes from the learner) are clampled with the mean value. As a result, ATD does not rely on any hyperparameters. We assume that the pairs that are easily identified by the discriminator are scored low due to the different action spaces, rather than a bad long-term strategy[1].

The imitation reward $r_t^{imt}$, which measures the similarity between the learner and the expert policies, depends on the method used for constructing the input. The full comparison of the imitation rewards is shown in Table 1. In all cases, we consider a scaling constant $0.5$, which makes the rewards positive when discriminator prediction is higher than $0.5$, meaning the discriminator is fooled. In practice, the rewards are clipped to be not larger than 10 to avoid numerical instabilities.

---

[1] We indeed observe this in our experiment. Short-term transitions are much more likely to obtain a low score when action spaces for the learner and the expert differs.

---

**Algorithm 1:** RILO training procedure

---

**input** : Set of expert trajectories $T_e$ and the coefficient $\lambda > 0$,
    Initial policy and discriminator parameters $\theta_0$ and $\phi_0$.
**output:** Policy $\pi_{\theta_K}$.
Initialize a success rate estimate $\upsilon_0$ to be 0.
**for** $k \leftarrow 1$ **to** $K$ **do**
 | Sample $\tau_k \sim Bernoulli(p = 1 - \upsilon_{k-1})$.
 | Get observations $\boldsymbol{l} = (s_0^l, ..., s_m^l) \sim \pi_{\theta k-1}(\tau_k)$ and environment rewards $\boldsymbol{r}^{env} = (r_1^{env}, ..., r_m^{env})$.
 | Update success rate estimate $\upsilon_k$.
 | **if** $\tau_k = 1$ **then**
  | Sample expert trajectory $\boldsymbol{e} = (s_0^e, s_1^e, ..., s_n^e) \sim T_e$.
  | Compute discriminator scores for expert and learner pairs of states:
    $\mathcal{D}_e = \{d_{i,j}^e = D_{\phi_{k-1}}(s_i^e, s_j^e) : i \neq j\}$   and   $\mathcal{D}_l = \{d_{i,j}^l = D_{\phi_{k-1}}(s_i^l, s_j^l) : i \neq j\}$.
  | Update $D_{\phi_k}$ to minimize $BCE(\mathcal{D}_e, 1) + BCE(\mathcal{D}_l, 0)$.
  | Build imitation rewards $\boldsymbol{r}^{imt} = (r_1^{imt}, ..., r_m^{imt})$, using $\mathcal{D}_l$.
  | Construct the final rewards $\boldsymbol{r} = \boldsymbol{r}^{enc} + \lambda \boldsymbol{r}^{imt}$.
 | **else**
  | Use environment rewards as the final rewards $\boldsymbol{r} = \boldsymbol{r}^{env}$.
 | Update $\pi_{\theta_k}$ with final rewards $\boldsymbol{r}$ and any RL-algorithm.

---

### 3.3 SELF-EXPLORATION

Since the action spaces between the two agents are not necessarily the same, it is unlikely that an optimal policy for the learner is the same as that for the expert. For example, imagine a situation in which expert action space is a subset of the learner action space, *i.e.* $\mathcal{A}^e \subset \mathcal{A}^l$ (*e.g.*, grid world in which the learner can move to all eight adjacent directions while the expert can only move on the four perpendicular directions). In this case, the learner can be penalized by the discriminator for performing actions in $\mathcal{A}^l \setminus \mathcal{A}^e$ (diagonal moves in the example) because it can easily be distinguished by the discriminator, even though those actions are optimal.

To resolve this issue, we propose to give the learner the possibility to explore the environment by being free from imitating expert's behaviour. As a result, the final form of Equation 2 becomes:

$$r_t = r_t^{env} + \lambda \cdot \tau_k \cdot r_t^{imt} , \quad \text{and} \quad \tau_k \sim \text{Bernoulli}(p = 1 - \upsilon_{k-1}), \tag{3}$$

where $\upsilon_{k-1}$ is estimated success rate. That is, the self-exploration parameter $\tau_k$ is a binary random variable controlling whether to consider $r_t^{imt}$ or not.

We set $\upsilon_k$ to be the estimate of the current learner policy success rate[2] and thus make the imitation reward guide the learning while the policy is not matured yet, *i.e.* during the early stage. As the behaviour of the learner becomes close to the expert trajectory, the success rate will increase accordingly and thus the policy can learn more with the guidance of environment rewards only. In other words, we want our agent to become independent of and not limited by the expert's supervision that may, as argued before, become harmful at some point during training. We empirically show that allowing the learner to interact with the environment without imitating leads to better results and the learner is more likely to use actions from $\mathcal{A}^l \setminus \mathcal{A}^e$. Importantly, the agent is aware of the value $\tau_k$, when it performs its actions which is realized by adding the binary feature to the learner input.

See the Algorithm 1 for the details of the training procedure.

## 4 EXPERIMENTS

In this section, we show that our approach performs favorably in the RILO setting. Our experiments aim to show that the agent can combine sparse, unshaped environmental reward with information from state-only observations (from the expert) to succeed in a navigation task. The specific questions we address are as follow. (1) Can we leverage sparse reward to improve over state-only IL methods? (2) How does the performance change when the action space of the expert and the learner are different? (3) Can the learner improves over the expert?

---

[2] In our experiments, we used a moving average to compute the success rate $\upsilon_k$.

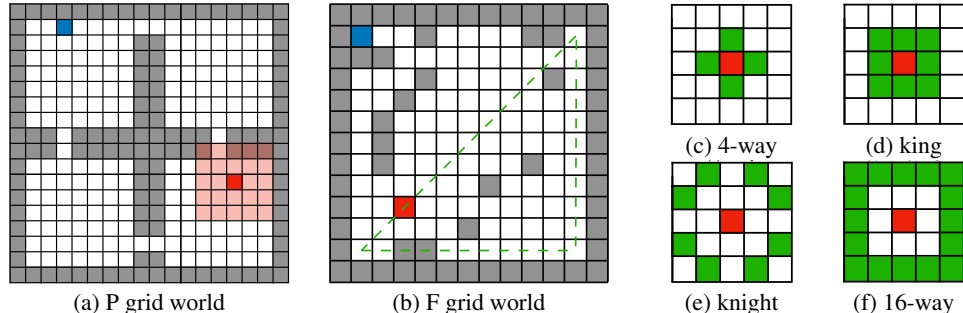

Figure 2: (a-b) Example of an initial state of the environment with blue and red squares representing the goal and the agent, respectively. For P grid world (a) the $5 \times 5$ visible area around the agent is highlighted. (c-f) Set of actions for different agents considered in our experiments. Green squares represent the possible locations after the move.

Due to limited space, we defer implementation details to Appendix A. Source code will be released upon acceptance.

## 4.1 EXPERIMENTAL SETUP

We conduct our experiment in two grid world environments: a fully observable grid world (we call it *F grid world*) and a partially observable one (*P grid world*). We consider the same set of simple navigation problems on both grid world environments. Both settings are designed in a way that a standard RL agent can succeed the tasks when the reward is dense and always fails when the reward is sparse.

**Environment** We consider two grid worlds with traps on the border (see Figure 2 (a-b)). At the beginning of each episode the goal is randomly (uniformly) located (for P map we restrict the target location to be one of the 4 corners). The agent's initial location depends on both the goal and the map. For the P environment, the agent is always placed in different room, and for F map the agent is placed in one of the locations covered by a triangle made out of the three left corners (a total of 30 possible initial locations per goal position – dashed lines on Figure 2 (b)). On F map each of the squares (not taken by the goal or the agent) has $15\%$ chances of being a trap (gray squares on figure) while for P grid world traps are fixed to create 4 rooms (although the passages are randomly placed for each episode). If any agent steps on a trap, the episode is terminated with a final reward of $-1$. The episode is also terminated, if the agent performs its $51^{th}$ (or $26^{th}$ for F map) action. All other rewards are zero unless the agent steps on the goal which gives reward of $+1$ and also terminates the game. In all experiments, we use discount factor $\gamma = 0.9$ and exploration rate $\epsilon = 0.05$. Note that the maps are different for each episode.

**Action spaces** Even though the grid environments are designed to show variability, the agents used on both maps have the same action spaces, which allows the simple comparison of results. In our setup, the action space of the agent is isomorphic to the set of its moves. We consider four move styles (as illustrated in Figure 2 (c-f)): *4-way* (up, down, left, right), *king* (like king in the chess game), *knight* (like knight in chess) and *16-way* (has to move by 2 in one direction (horizontal or vertical), and 0, 1 or 2 in another one)[3]. Note that the action space of 4-way and king agents (always move to adjacent locations) are disjoint from the other two agents (never move to adjacent squares). Also, the actions spaces of 4-way and knight are subsets of king and 16-way action spaces, respectively.

**Experts** We consider two different types of rewards, that we call sparse and dense. The sparse version was described before and provides a signal at the termination of each episode only, giving either a reward $+1$ or $-1$ in case of success or failure of the task, respectively. As purposely designed, no agent is able to achieve the goal using the sparse reward function. We engineered the dense rewards such that all agents are able to succeed in this task and the four experts are trained in this way for each map.

---

[3] Knight and 16-way are able to "jump" over traps in F map.

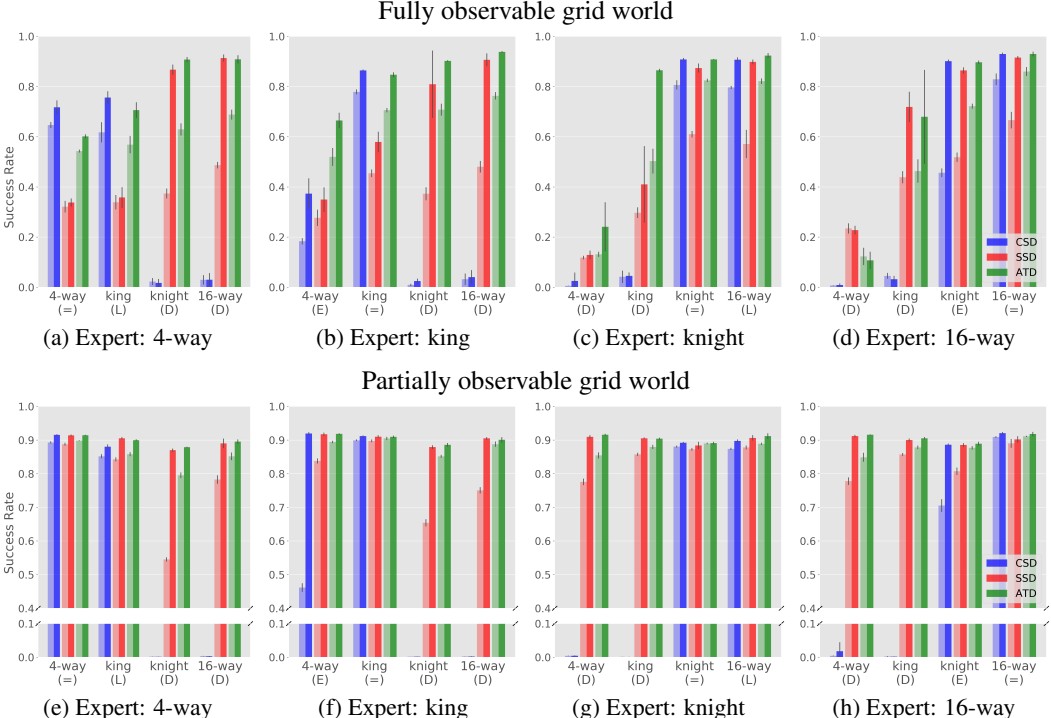

Figure 3: Comparison of RILO experiments with different imitation rewards and the effect of self-exploration on different expert-learner pairs. In the first row results for F map are presented (a-d) while results for P grid world are shown in the second row (e-h). Each chart represents how a given expert aids each learner, *i.e.* learner move styles vary while expert is fixed. We show results in which agents use self-exploration (right, dark) or not (left, bright). Additionally, the relation between the learner and the expert is coded using one of four symbols: the same action spaces (=), disjoint action spaces (D), superior learner (L), or superior expert (E).

**Experiments** For each map we consider all possible expert-learner agent combinations, resulting in a total of 16 pairs. For each pair, we compare all methods proposed in Section 3. Each experiment is executed five times (with different seeds) and results are shown with their mean and standard deviation over all trials. We consider thousands of observations (expert trajectories) for P map, and ten thousands of them for F grid world. This number is about three order of magnitude lower than the number of trails required to train the expert agents with dense reward. We note that maps are randomly built for each episode and hence, just a fraction of maps have the expert observation performed on them.

## 4.2 EXPERIMENTAL RESULTS AND DISCUSSIONS

We compare methods with different imitation reward strategies. Figure 3 shows results for all possible 16 expert-learner pairs considering three different imitation rewards: SSD, CSD and ATD. We consider the first two: CSD (a method similar to Torabi et al. (2018b)), and SSD (similar to Merel et al. (2017)), as baseline methods. To disentangle the effect of the self-exploration mode, we also show results where learners are trained with self-exploration (right, dark bar on the figure), dubbed SSD-SE, CSD-SE and ATD-SE, or not (left, bright bar on the figure).

We notice that self-exploration significantly helps in the RILO setting. The learner policy is consistently better when given the opportunity to explore the environment without expert supervision. In very simple cases (for example the same action spaces for the expert and the learner), the effect of self-exploration usually reduces to a few percent points, although it still helps significantly.

**Same action spaces** When the move styles are the same for the learner and the expert, $\mathcal{A}^e = \mathcal{A}^l$ (code (=) on Fig. 3), the performance of CSD and ATD (and their counterparts with self-exploration) are similar. It means that using the consecutive states works well when the learner and expert share the same action space. SSD, the second baseline, works well in same action spaces cases on P map,

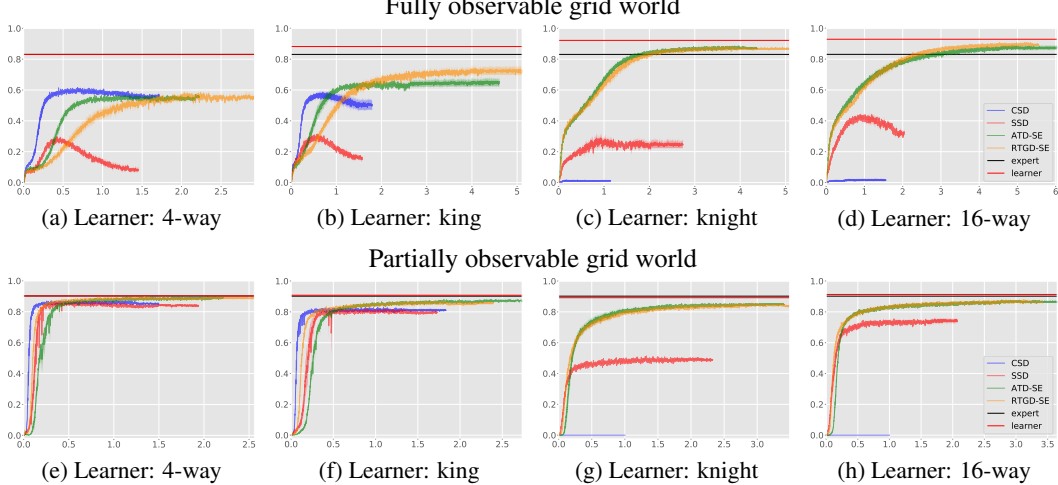

Figure 4: Performance of algorithms (success rate) with respect to the number of iterations (in millions), assuming different learners. In all cases, the expert is the 4-way agent. See text for details and Appendix B for higher resolution figures or other experts results.

but it is significantly worse on F map. However, the difference tends to diminish when the agent is trained in self-exploration mode (SSD-SE).

**Disjoint action spaces** Consecutive states should not be used when learner and expert have disjoint actions spaces, $\mathcal{A}^e \cap \mathcal{A}^l = \emptyset$ (code (D) on Figure 3). In all these cases (eight of them for each environment), CTD is not able to solve the task and the final success rate never exceeds 5%. Even self-exploration is not able to give any improvement due to the small success rate. On the other hand, with the use of self-exploration, SSD-SE and ATD-SE perform very well in disjoint cases, obtaining the best success rates.

**Superior learner** Next, we analyze the scenario where the learner has superior set of actions, *i.e.* $\mathcal{A}^e \subsetneq \mathcal{A}^l$ (code (L) on Figure 3). It is, of course, not disjoint case and then all methods perform well. However, SSD and SSD-SE perform significantly worse on F map. Note that the superior learner can always imitate the expert (and limit itself to smaller number of actions) so discriminator is not able to observe that different actions may be performed. We noticed, however, that the self-exploration agents achieve the goal faster (in terms of number of steps) in this scenario. Hence, the learner is more likely to use actions that cannot be used by the expert but lead to better solution (note that due to the discount factor the solutions using less steps are preferable). In these four (two per environment) learner-expert scenarios, the learner achieve the goal in about 15% less steps when trained with self-exploration (when CSD-SE or ATD-SE used).

**Further comparison to baselines** Figure 4 compares the methods when 4-way expert is used along with all different learners (for both F grid world and P grid world). Results for other experts can be found in the Appendix B. We present agents ATD-SE and RTGD-SE (our methods) and the baseline methods (CSD and SSD). We also included the performance of both agents trained with dense reward. We noticed in our preliminary experiments that RTGD with minimal gap set to three tends to work the best and hence we fix that hyper-parameter for all our experiments.

As shown in Figure 4, when expert and learner actions are disjoint (c-d, g-h) our methods usually reach significantly better performance and converge much faster. In these cases for F environment, our methods are able to outperform the expert, approaching the upper bound of the learner trained with dense reward. Again, the CSD performs well only when the learner actions are the same or are superset of expert actions (a-b, e-f). We can also see the signs of overfitting for SSD strategy trained on F map.

**Observation limit** The same set of 16 experiments were performed with smaller amount of expert observation trajectories. When 500 trajectories are used for P environment (originally 1000 observations) the results just slightly deteriorate. When this number is further limited to 100 trajectories only, the positive effect of self-exploration is even more apparent. However, this reduction makes

our problem much more challenging and only 12 out of 16 methods remain solved (by at least one approach). Using 50 trajectories is enough for only a few expert-learner pairs.

We also checked how different observation limits influence results on F map (originally ten thousand of observations). We considered only one thousand of expert observations. This reduction turns out to make our problem very challenging. Baseline methods (CSD and SSD) never bypass 50% success rate, and usually score around 20%. ATG and RTGD perform significantly better: a success rate above 50% in 8 out of 16 expert-learner pairs (but achieves more than 90% only in three cases). When self-exploration used, both methods (ATG-SE and RTGD-SE) achieve success rate above 90% in 8 out of the 16 setups. We note that with smaller number of observations RTGD performs slightly better than ATD, probably due to the fact that is is harder to overfit discriminator when the inputted pairs are random.

At the first glance, it may be surprising that fully observable grid world requires more observations (and also the agents usually perform better in P map as compared to F map). However, the access to the full map and stochasticity of the maps make it much easier for discriminator to remember maps and discriminate based on that. Hence, we believe that while fully observability makes the problem easier for the agent, the discriminator "benefits" more and that makes the training procedure loop more challenging.

**Coherence of results**    Results on both maps are coherent and justify the importance of self-exploration. The only significant difference in results is that the gap between SSD-SE and ATD-SE (or RTGD-SE) is smaller for P grid world. We hypothesize that this is due to the fact that the discriminator is more prone to overfitting (remember all expert trajectories) when given the full map (not just a small part as in P environment) and that problem is more severe when only a single state is given, not a random pair of states. To validate this hypothesis we train the models assuming unlimited observations from the expert for F map. In this case, the performance of SSD-SE approaches that of ATD-SE and RTGD-SE. It means that when given the massive amount of expert data, simply having the distribution of states is enough to infer the policy.

**Comparison to pure IL**    We also checked how the presented strategies work in a pure IL setting, *i.e.* when the sparse environment reward is not given. Not surprisingly, the performance of all method deteriorate. Our methods (RTGD and ATD) again perform better than the baseline methods. However, in pure IL setup the self-exploration cannot be applied and then the differences are smaller.

## 5 CONCLUSION

In this paper, we show that by leveraging unshaped rewards from the environment, an agent is able to outperform standard state-only imitation learning. Our proposed method efficiently combines the sparse environment rewards with the standard imitation learning objective. We show experimentally that this approach achieves good performance over baselines in the RILO setting. Our method is especially well-suited when the actions of the trained agent differ from those of the expert. We also show that an agent trained with our approach can outperform the expert by using the sparse rewards in an optimized way.

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

APPENDIX

## A   IMPLEMENTATION DETAILS

As mentioned in Subsection 3.1, our method is composed of two trainable components, a policy network $\pi_\theta : \mathcal{S} \to \mathcal{A}^l$ and a discriminator $D_\phi : \mathcal{S} \times \mathcal{S} \to [0, 1]$. Both functions are parameterized by neural networks, having a state encoder that has identical architectures but with different set of weights. All weights are optimized using Adam (Kingma & Ba, 2014) with a learning rate of $10^{-4}$. The training procedure is terminated after one million of episodes without significant performance improvement (defined to be $1\%$ average success rate increase).

**State encoder**   The encoder receives as input the grid world encoded as a matrix (for P map only a $5 \times 5$ subgrid around the agent is taken while the full $13 \times 13$ grid is visible in F environment) with 4 possible values: 3 for a goal, 2 for an agent, $-1$ for all traps and walls and 0 otherwise) and a few additional features[4]. The map is processed by 5-layer CNN with kernel size 3 and residual connections. Then, it is flattened and concatenated with additional features which constitutes the final state encoding.

**Policy network**   We use (synchronous) advantage actor-critic (A2C) algorithm (Mnih et al., 2016) to optmize the policy. The policy network encodes the state and then the final transformation is applied to obtain $(k + 1)$ dimensional vector ($k$ possible actions, modeled as a probability with softmax, and one dimension to represent the value-function). The type of the final transformation depends on the environment. It is fully connected (with ReLU) for fully observable grid world and LSTM for partially observable grid world (since the memory is needed to perform the task well).

**Discriminator**   The discriminator encodes both input states separately (using the same state encoder) that are next separately inputted to 2-layer MLP with 256 hidden units (both layers) and ReLU activation function. The difference of the two encoded states are then fed to two fully-connected layers, with outputs sized 256 and 1, respectively. The final output is transformed into a probability with a sigmoid function.

Note that the computation time for the final state tuple is negligible compared to the computation for state encoding. As a result, ATD does not carries a heavy computation burden in case of relatively small trajectories. In case of very long trajectories, however, a fixed number of random pairs should be considered.

## B   ADDITIONAL PLOTS

Similar to Figure 4, we show performance of all algorithms with respect to the number of iterations on Figures 5-8. Each plot assumes different expert (see caption) and presents result for all four learners.

---

[4]These features are two floats $(x, y) \in (0, 1)^2$ encoding agent position. Additionally, as described before, the learner trained with self-exploration is given a binary variable indicating the nature of the reward that will be used.

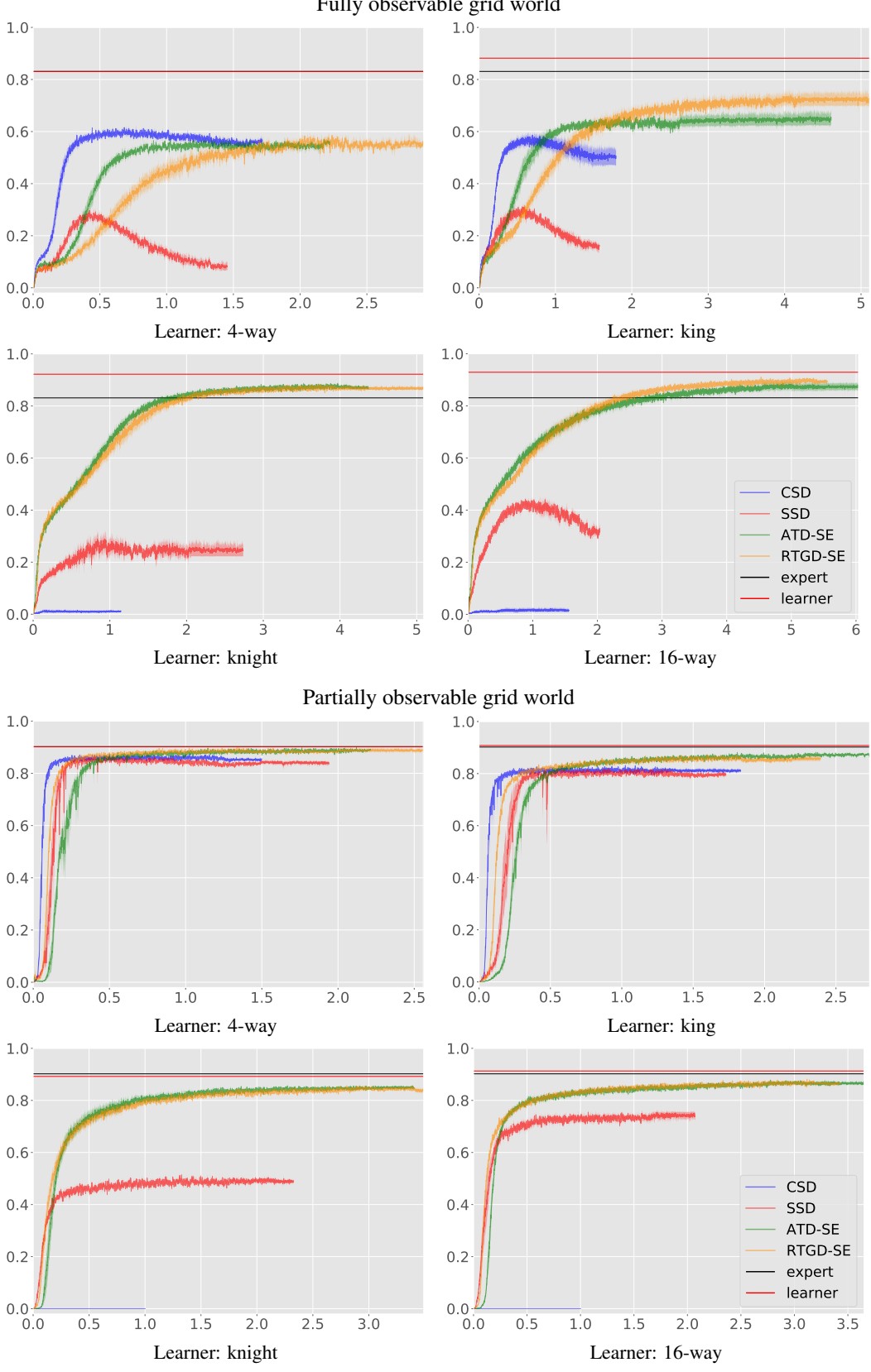

Figure 5: Expert: 4-way

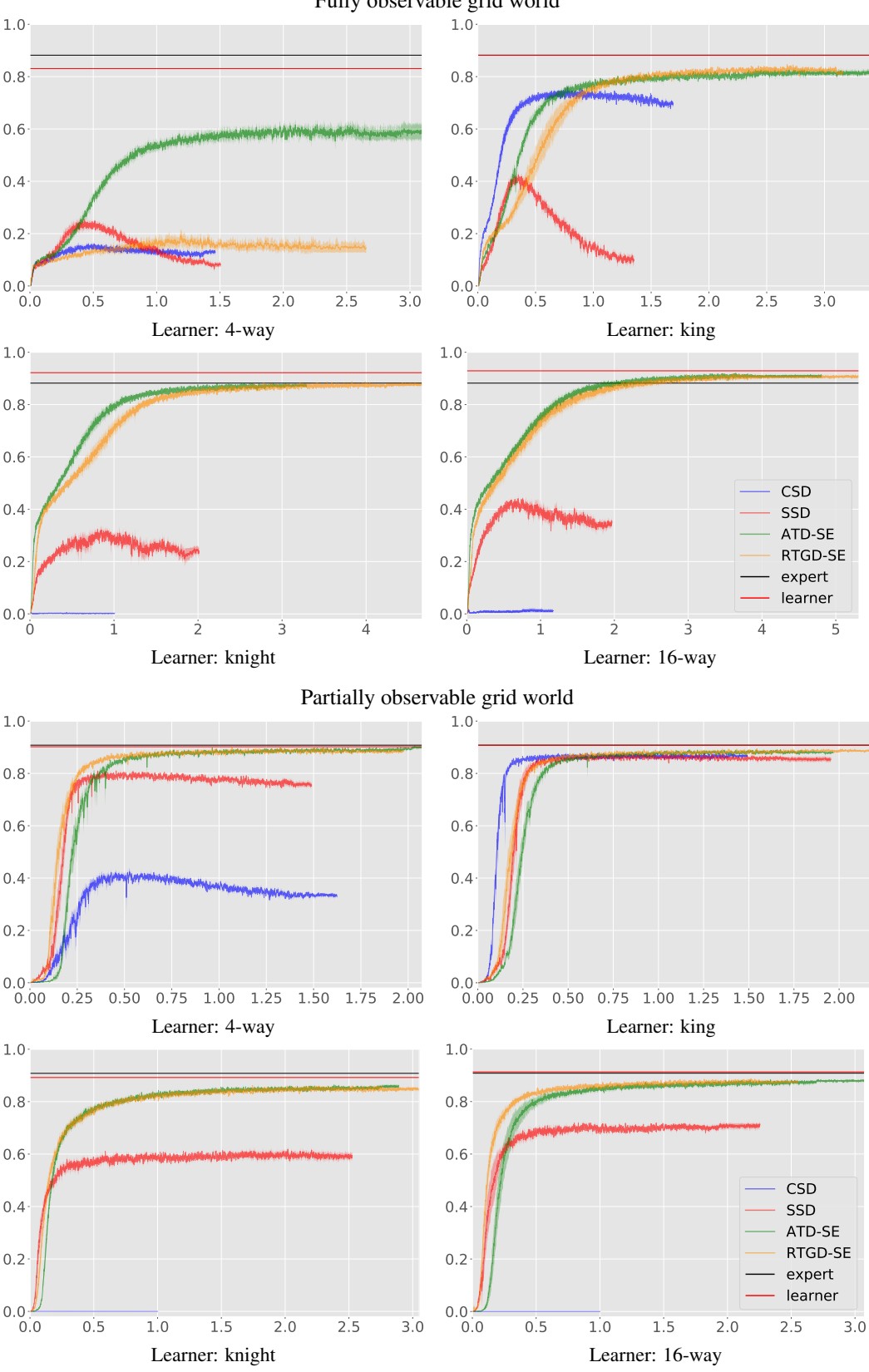

Figure 6: Expert: king

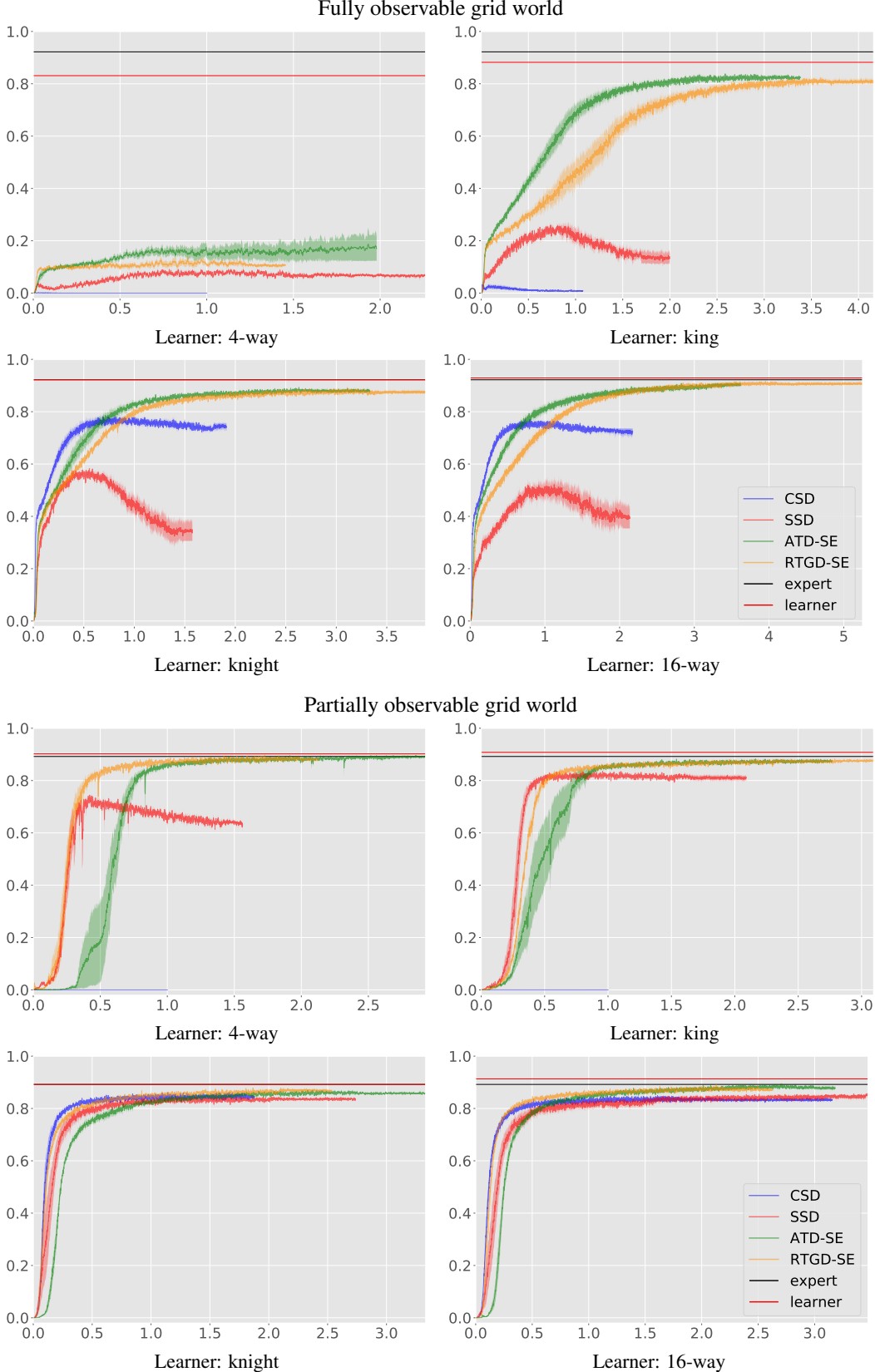

Figure 7: Expert: knight

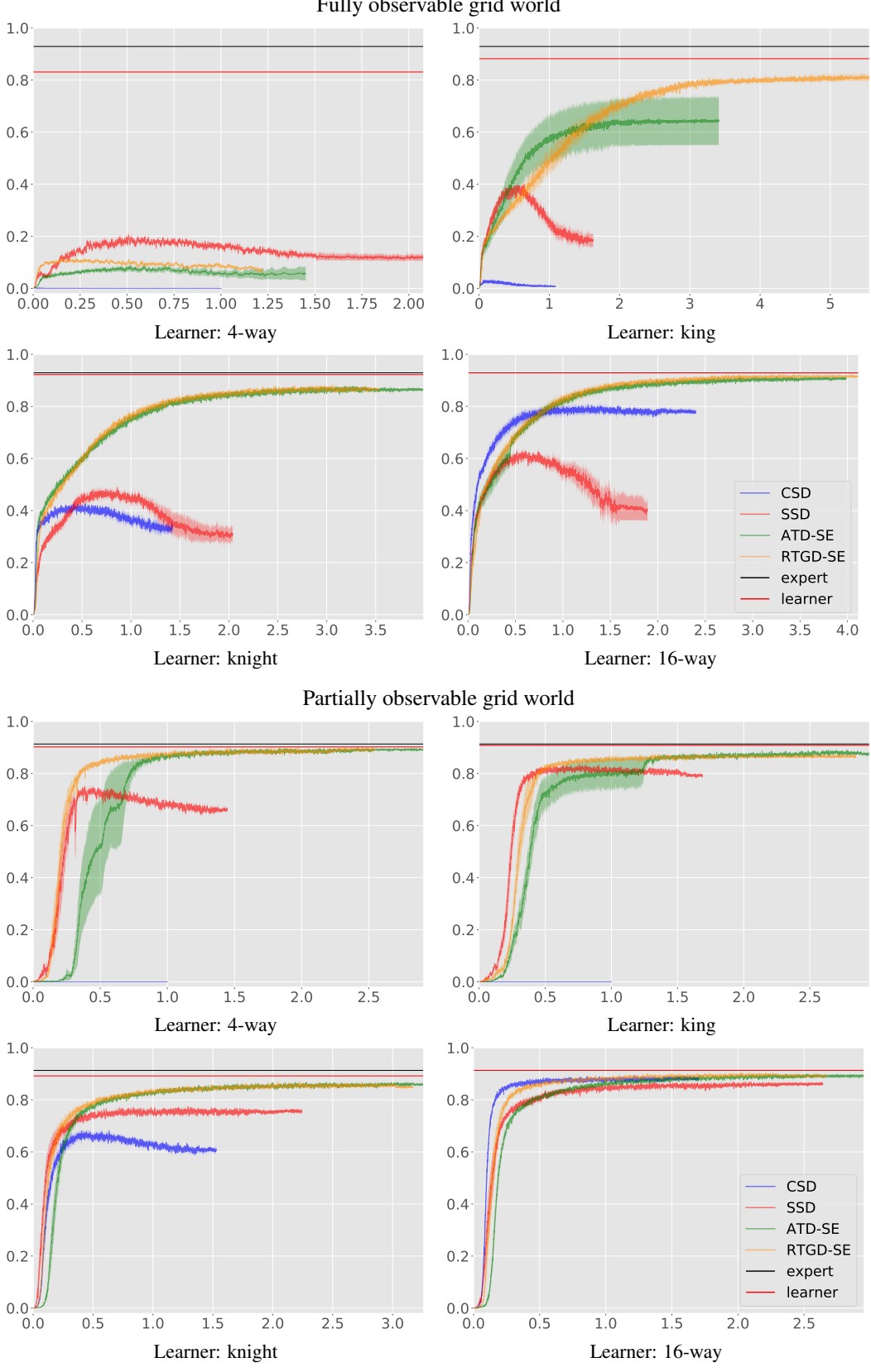

Figure 8: Expert: 16-way

