# OpenReview forum: "Reinforced Imitation Learning from Observations"
_ICLR.cc/2019/Conference_

### Official Review · AnonReviewer1 · 2018-11-03
**First review**

**Rating:** 4
**Confidence:** 4

**Review:**

This paper proposes some new angles to the problem of imitation learning from state only observations (not state-action pairs which are more expensive).
Specifically, the paper proposes "self exploration", in which it mixes the imitation reward with environment reward from the MDP itself in a gradual manner, guided by the rate of learning.
It also proposes a couple of variants of imitation rewards, RTGD and ATD inparticular, which formulate the imitation rewards for random or exhaustive pairs of states in the observation data, as opposed to the rewards proposed in existing works (CSD, SSD), which are based on either consecutive or single states, which constitute the baseline methods for comparison.
The authors then perform a systematic experiment using a particular navigation problem on a grid world, and inspect under what scenarios (e.g. when the action spaces of the expert and learner are the same, disjoint or in a containment relationship) which of the methods perform well relative to the baselines.
Some moderately interesting observations are reported, which largely confirm one's intuition about when these methods may perform relatively well.
There is not very much theoretical support for the proposed methods per se, the paper is mostly an empirical study on these competing reward schemes for imitation learning.
The empirical evaluation is done in a single domain/problem, and in that sense it is questionable how far the observed trends on the relative performance of the competing methods generalizes to other problems and domains.
Also the proposed ideas are all reasonable but relatively simple and unsurprising, casting some doubt as to the extent to which the paper contributes to the state of understanding of this area of research.

---

> ### Author Response · Authors · 2018-11-08
> **Initial reply**
>
> We thank you for the constructive comments.
> We agree that these results corroborates with one's intuition. And this is precisely why we think the results are interesting.
>
> Although the experiments show the main properties of the proposed algorithm, we also agree that more experiments on different tasks would definitely be helpful.
> We explored the method in another gridworld setting, in which the agent does not have access to the whole map, but only partial observations (we use a 5x5 subgrid surrounding the agent). We used the same action spaces for both expert and learner to be able to compare the result directly. Results in the POMDP setting are very similar to the fully-observed setting.
> We started experimenting on more complex environment, Vizdoom (3D POMPD). This environment is much more (computationally) demanding, so conducting a systematic set of  experiment (as done for the original case) is more challenging. Preliminary results lead to the similar conclusions, but we need more time to make any statement in this setting.
>
> We would appreciate to hear your opinion about choices for additional experiments and suggestion on what we can add to the paper to make it better.

---

### Official Review · AnonReviewer2 · 2018-11-04
**State only demonstrations but in deterministic environments**

**Rating:** 5
**Confidence:** 5

**Review:**

The paper proposes to combine expert demonstration together with reinforcement learning to speed up learning of control policies. To do so, the authors modify the GAIL algorithm and create a composite reward function as a linear combination of the extrinsic reward and the imitation reward. They test their approach on several toy problems (small grid worlds).

The idea of combining GAIL reward and extrinsic reward is not really new and quite straight forward so I wouldn't consider this as a contribution. Also, using state only demonstration in the framework of GAIL is not new as the authors also acknowledge in the paper. Finally, I don't think the experiments are convincing since the chosen problems are rather simple.

But my main concern is that the major claim of the authors is that they don't use expert actions as input to their algorithm, but only sequences of states. Yet they test their algorithm on deterministic environments. In such a case, two consecutive states kind of encode the action and all the information is there. Even if the action sets are different in some of the experiments, they are still very close to each other and the encoding of the expert actions in the state sequence is probably helping a lot. So I would like to see how this method works in stochastic environments.

---

> ### Author Response · Authors · 2018-11-08
> **Initial reply**
>
> We thank the reviewer for the detailed comments.
>
> We fully agree (as acknowledged that in the paper) that using state-only demonstration in the GAIL framework is not new.  However, to the best of our knowledge, no systematic experiments studying state-only demonstrations have been done. We propose baseline methods that are straightforward implication of previous works and we compare them to our methods (RTGD, ATD, self-exploration). We show quantitatively how all methods presented perform in different situations.
>
> We agree to the fact that in deterministic environments the consecutive states encode the action performed. The choice of deterministic environment was made on purpose. We believe that this choice makes the problem more challenging because discriminator is more likely to ‘decode’ action spaces used by two agents and easily discriminate based on that -- which leads to providing non-informative rewards (rewards that are not linked with the policy).
> This intuition is confirmed in the paper: CSD performs bad when action spaces are disjoint. We report results for methods that can not decode actions (RTGD, ATD and SSD) and they perform better than CSD (unless action states are not disjoint, then all methods perform similar). We will modify the paper to make this more clear.
>
> Although the experiments show the main properties of the proposed algorithm, we also agree that more experiments on different tasks would definitely be helpful.
> We explored the method in another gridworld setting, in which the agent does not have access to the whole map, but only partial observations (we use a 5x5 subgrid surrounding the agent). We used the same action spaces for both expert and learner to be able to compare the result directly. Results in the POMDP setting are very similar to the fully-observed setting.
> We started experimenting on more complex environment, Vizdoom (3D POMPD). This environment is much more (computationally) demanding, so conducting systematic experiment set (as done for the original problem) is more challenging. Preliminary results lead to the similar conclusions, but we need more time to make any statement in this setting.
> Vizdoom environment dynamic implements inertia and hence the previous action cannot be inferred from consecutive states. Also the behavior of another agents (bots) is non-deterministic.
>
> We would appreciate any suggestions on the experimental setting that could improve our work.

---

### Official Review · AnonReviewer3 · 2018-11-06
**heuristic combining environment rewards with an IRL-style rewards**

**Rating:** 6
**Confidence:** 2

**Review:**

The draft proposes a heuristic combining environment rewards with an IRL-style rewards recovered from expert demonstrations, seeking to extend the GAIL approach to IRL to the case of mismatching action spaces between the expert and the learner. The interesting contribution is, in my opinion, the self-exploration parameter that reduces the reliance of learning on demonstrations once they have been learned sufficiently well.

Questions:

- In general, it's known that behavioral cloning, of which this work seem to be an example in so much it learns state distributions that are indistinguishable from the expert ones, can fail spectacularly because of the distribution shift (Kaariainen@ALW06, Ross&Bagnell@AISTATS10, Ross&Bagnell@AISTATS11). Can you comment if GAN-based methods are immune or susceptible to this?

- Would this work for tasks where the state-space has to be learned together with the policy? E.g. image captioning tasks or Atari games.

- Is it possible to quantify the ease of learning or the frequency of use of the "new" actions, i.e. $A^l \setminus A^e$?. Won't learning these actions effectively be as difficult as RL with sparse rewards? Say, in a grid world where 4-way diagonal moves allow reaching the goal faster, learner is king 8-way, demonstrations come from a 4-way expert, rewards are sparse and each step receives a -1 reward and the final goal is large positive -- does the learner's final policy actually use the diagonals and when?

Related work:

- Is it possible to make a connection to (data or policy) aggregation methods in IL. Such methods (e.g. Chang et al.@ICML15) can also sometimes learn policies better than the expert.

Experiments:
- why GAIL wasn't evaluated in Fig. 3 and Fig. 4?

Minor:
- what's BCE in algorithm 1?
- Fig.1: "the the"
- sec 3.2: but avoid -> but avoids
- sec 3.2: be to considers -> be to consider
- sec 3.2: any hyperparameter -> any hyperparameters
- colors in Fig 2 are indistinguishable
- Table 1: headers saying which method is prior work and which is contribution would be helpful
- Fig. 3: if possible try to find a way of communicating the relation of action spaces between expert and learner (e.g. a subset of/superset of). Using the same figure to depict self-exploration make it complicated to analyse.
- sec 3.2: wording in the last paragraph on p.4 (positive scaling won't _make_ anything positive if it wasn't before)

---

> ### Author Response · Authors · 2018-11-10
> **Initial reply**
>
> Thanks for the review and the feedback.
> We are especially thankful for raising the topic of exploring the "new" actions, i.e. $A^l \setminus A^e$. Our method, in contrary to previously presented, does not limit the learner to use the same actions as used in expert demonstrations. As shown in our experiments, we were able to successfully train our agent even when action spaces are disjoint. Considering your specific example (with 8-way-king learner and 4-way expert), we already have a short comment on that in the paper -- the learner on average gets to the goal quicker (in respect to number of steps), hence it uses “new” actions. We believe that it is important remark and we will edit paper to make this more clear and visible.
> We did not have any experiments supporting the hypothesis of being immune to domain shift in our setup. That’s a very interesting direction, but we believe it is general question for all GAN-based methods and should be a part of another work. However, we would be glad to explore this in future research.
> I believe that our idea will work in any setup but it is well-suited for cases when expert and learner have different action spaces. The Atari games are created in a such way that all buttons (actions) are important and usually necessary to perform well. However, we started experimenting on complex environment, Vizdoom (3D POMPD), where the agent is still able to perform well using just a subset of all possible actions. Preliminary results lead to the similar conclusions, but we need more time to make any statement in this setting.
> We have not shown GAIL for comparison in Fig 3. and Fig. 4 because the straightforward application of GAIL method is possible only when learner action space is a subset of expert action space which is rarely case in our experiments.
> Thanks you again for providing related work and all minor remarks. We will update our paper according to them.

---

### Public Comment · (anonymous) · 2018-11-18
**Some comment on novelty**

I think this paper basically makes a incremental follow-up to [Kang et.al., 2018] and [Li et.al., 2017]. However, it seems that the authors do not show enough respect for these work. From the perspective of [Kang et.al., 2018], this paper simply changes GAIL over (s, a) into GAIL over (s_t,.. s_t+n), which has been repeatedly proposed in [Bradly et.al., 2016][Josh et.al., 2017][Faraz et.al., 2018]. On the other hand, from [Li ei.al., 2017], we could also consider this paper as a variant of reward augmentation trick.

Overall, I think the novelty of this paper is quite limited and may not fit into ICLR requirements.

[Kang et.al., 2018] Policy Optimization with Demonstrations, Kang et.al., in ICML, 2018
[Li et.al., 2017] InfoGAIL: Interpretable Imitation Learning from Visual Demonstrations, Li et.al., in NIPS, 2017
[Bradly et.al., 2016] Third-Person Imitation Learning, Bradly et.al., in ICLR, 2017
[Josh et.al., 2017] Learning human behaviors from motion capture by adversarial imitation, Josh et.al., arXiv 1707.02201
[Faraz et.al., 2018] Generative Adversarial Imitation from Observation, Faraz et.al., arXiv 1807.06158

---

> ### Author Response · Authors · 2018-11-20
> **Response**
>
> We thank you for your feedback.
> We agree that three of the references mentioned are very relevant and they are highly cited throughout the paper (they are also baselines which we use in experimental section). Hence, we believe that we show enough respect for these work.
>
> We fully agree (and acknowledge in the paper) that using state-only demonstration in GAIL framework is not a novelty. However, to the best of our knowledge, no systematic experiments studying state-only demonstrations have been done. Especially for the case of different action spaces.
> We include baselines that are straightforward implication of previous work (the ones aforementioned and dully cited on the paper) and we compare them to what we propose (RTGD, ATD, self-exploration). We show quantitatively how learning is achieved under different teacher-student situations.
>
> Third-Person Imitation Learning focus mostly on the domain adaptation viewpoint issue, while assuming same action spaces between student and teacher. However, we understand this is an important application of GAIL and will mention it in the Introduction.
>
> The reward augmentation trick in InfoGAIL adds surrogate reward with a fixed weight for the whole training procedure. We do not agree that self-exploration is a variant of that since in our case the weight (zero or one) dynamically changes and is a part of an agent input. The environment reward is also assumed to be sparse and in our experiments it is always added only once, in the end of the trail (the last environment reward).
> Our understanding of InfoGAIL makes us believe that the work focuses on different aspect of imitation learning. However, similar to Third-Person Imitation Learning work, InfoGAIL is a  well-known paper that uses GAIL and we will consider mentioning that.

---

### Author Response · Authors · 2018-11-26
**Paper update**

We would like to thank for all reviews again. We updated the paper based on them.

The main update is the description of (and the results on) the partially observable environment. We wanted to design the new environment to be different from the one presented while being able to perform the same set of experiments to check that our findings are general.

We decided to use another grid world with agents having same action spaces considered before. However, the new environment differs a lot. First of all, it is partially observable and just a small subgrid (5x5) around the agent is inputted to the models (both the policy and the discriminator). The map is larger and consist of four small rooms with randomly located passages between them. Hence, the agent has to explore and get to know the map to reach the goal. That makes the use of LSTM for policy necessary.

Despite all these differences, the results for the new environment are coherent with the previous results and justify the importance of self-exploration. They further confirm our intuition and provide the answer which of the discriminator inputs should be used based on the relation between the expert and the learner action spaces.

Although the experiments conducted in the first version of the paper show the main properties of the proposed algorithm, we agree that the extra experiments added in the new version are important since the methods should generalise to different problems. We hope that the new set of experiments address the main concerns of the reviewers. Additionally, we have cleaned our source code that will be released upon acceptance, so our results are easily reproducible.

---

### Meta-Review · Area_Chair1 · 2018-12-15

**Confidence:** 4
**Recommendation:** Reject

**Metareview:**

This paper proposes to combine rewards obtained through IRL from rewards coming from the environment, and evaluate the algorithm on grid world environments. The problem setting is important and of interest to the ICLR community. While the revised paper addresses the concerns about the lack of a stochastic environment problem, the reviewers still have major concerns regarding the novelty and significance of the algorithmic contribution, as well as the limited complexity of the experimental domains. As such, the paper does not meet the bar for publication at ICLR.